# Effect of RONS-Induced Intracellular Redox Homeostasis in 6-NBDG/Glucose Uptake in C2C12 Myotubes and Single Isolated Skeletal Muscle Fibres

**DOI:** 10.3390/ijms24098082

**Published:** 2023-04-29

**Authors:** Escarlata Fernández-Puente, Eva Martín-Prieto, Carlos Manuel Márquez, Jesús Palomero

**Affiliations:** 1Department of Physiology and Pharmacology, University of Salamanca, 37007 Salamanca, Spain; 2Institute of Neurosciences of Castilla y León (INCyL), 37007 Salamanca, Spain; 3Institute of Biomedical Research of Salamanca (IBSAL), 37007 Salamanca, Spain

**Keywords:** glucose uptake, 6-NBDG, insulin resistance, C2C12 myotubes, skeletal muscle fibres, ROS, hydrogen peroxide, nitric oxide, redox homeostasis, quantitative fluorescence microscopy

## Abstract

The glucose uptake in skeletal muscle is essential to produce energy through ATP, which is needed by this organ to maintain vital functions. The impairment of glucose uptake compromises the metabolism and function of skeletal muscle and other organs and is a feature of diabetes, obesity, and ageing. There is a need for research to uncover the mechanisms involved in the impairment of glucose uptake in skeletal muscle. In this study, we adapted, developed, optimised, and validated a methodology based on the fluorescence glucose analogue 6-NBDG, combined with a quantitative fluorescence microscopy image analysis, to determine the glucose uptake in two models of skeletal muscle cells: C2C12 myotubes and single fibres isolated from muscle. It was proposed that reactive oxygen and nitrogen species (RONS) and redox homeostasis play an important role in the modulation of intracellular redox signalling pathways associated with glucose uptake. In this study, we prove that the prooxidative intracellular redox environment under oxidative eustress produced by RONS such as hydrogen peroxide and nitric oxide improves glucose uptake in skeletal muscle cells. However, when oxidation is excessive, oxidative distress occurs, and cellular viability is compromised, although there might be an increase in the glucose uptake. Based on the results of this study, the determination of 6-NBDG/glucose uptake in myotubes and skeletal muscle cells is feasible, validated, and will contribute to improve future research.

## 1. Introduction

Skeletal muscle is the largest organ in the body and exerts different functions such as locomotion, equilibrium, heat generation, and metabolism, which involve a continuous demand for and consumption of energy [1,2]. ATP is the key molecule that transfers energy through biochemical reactions that take place in the cell. The main substrate that produces ATP is glucose, and the breakdown of glucose led by the biochemical reactions of anaerobic and aerobic glycolysis generates ATP in the cell [3]. Therefore, cells from tissues with high demand for energy, such as skeletal muscle, need a continuous supply of glucose to guarantee the production and disposal of ATP [3]. Before ATP generation, glucose must be internalised into the cell from the extracellular space. This process is called glucose uptake [4]. Glucose uptake in skeletal muscle is mainly regulated by insulin. This hormone, insulin, binds to specific receptors located in the extracellular face of the plasma membrane of skeletal muscle cells. Insulin binding activates an intracellular pathway that drives the translocation of the glucose transporter GLUT4 from the cytosol to the plasma membrane, which leads to the internalisation of glucose from the extracellular space to the cytosol [4]. In this way, the supply and disposal of glucose to the cell is secured for ATP production [3,4]. When glucose uptake is dysregulated or impaired, the intracellular disposal of glucose may be compromised and thus affect the production of energy and cellular metabolism, which lead to sequential pathological states ending with the onset of diabetes type 2 and the negative consequences and symptoms that characterise this disease [5,6].

Skeletal muscle fibres are the main type of cells that constitute skeletal muscle and are postmitotic multinuclear cells formed from the fusion of single cells, myoblasts, which differentiate into myotubes and finally mature to become fibres [7,8]. As mentioned above, insulin leads the main physiological control of glucose uptake in skeletal muscle. However, it has been proposed that other molecules associated with intracellular redox signalling (e.g., reactive oxygen and nitrogen species (RONS)) might be involved in the regulation of glucose uptake in skeletal muscle [9,10,11]. Therefore, there is an expectation in the field of redox biology to investigate the role of RONS in the process of glucose uptake in skeletal muscle. Knowledge in this area could help find new targets to treat the impairment of glucose uptake and insulin resistance manifested in type 2 diabetes, obesity, and ageing [6,12].

Glucose uptake in skeletal muscle has been studied using different experimental models in vivo, in vitro, and ex vivo, focusing on biological samples such as whole muscle, muscle biopsies, and cellular models. Furthermore, the analytical determination of glucose uptake has been carried out using different analytical techniques. The most frequent include biochemical techniques, such as those associated with the enzymatic activity of glucose oxidase [13], radioisotopic methodologies using a radioactive glucose analogue, 2-deoxy-D [1,2-^3^H] glucose [14,15], nuclear magnetic resonance imaging [13], and fluorescence imaging to monitor the fluorescence glucose analogue 2-NBDG (2-(N-(7-Nitrobenz-2-oxa-1,3-diazol-4-yl)Amino)-2-Deoxyglucose) [14,15,16,17]. As in every field of scientific research, the choice of experimental model and analytic technique depend on the aims of the research, methodology, and resources available. However, it is crucial for the success of research to be aware of the advantages and disadvantages of experimental models and analytical techniques that will be implemented and incorporating proper adaptations, if they are feasible and suitable, to the model(s) that are planned to be used in the research.

In this study, we implement suitable and feasible methodological approaches that will provide a robust tool to determine glucose uptake in skeletal muscle. We determine the glucose uptake in two models of skeletal muscle. One is the C2C12 myotube model, whose cells are obtained from the fusion (differentiation) of C2C12 myoblasts (a mouse skeletal muscle cell line), and the other is a model of single mature skeletal muscle fibres isolated from the mouse flexor digitorum brevis muscle. The analytical technique used to determine the glucose uptake in our models is based in the use of a fluorescence glucose analogue, 6-NBDG (6-(N-(7-Nitrobenz-2-oxa-1,3-diazol-4-yl)amino)-6-Deoxyglucose). 6-NBDG is a fluorescence nonhydrolyzable glucose analogue that is transported across the plasma membrane by the glucose transporter GLUT4 in skeletal muscle and retained in the cell without any transformation or metabolisation [18,19]. This feature is essential to determine glucose uptake. In combination, cells (myotubes and fibres) are monitored by quantitative fluorescence microscopy imaging in live cells to register the intracellular presence of 6-NBDG. In this way, the quantitative fluorescence data from the cells represent 6-NBDG that is incorporated into the cytosol, which can be extrapolated as a glucose uptake by the cell. 6-NBDG has been used to quantify the glucose uptake via confocal fluorescence microscopy in astrocyte cultures [19,20], in monocytes populations using flow cytometry [21], in 3T3-L1 adipocyte cultures using a fluorescence microplate reader [18], and in L6 myotube cultures using a fluorescence microplate reader [22]. However, to the best of our knowledge, this is the first time that 6-NBDG was used to determine the glucose uptake specifically in single C2C12 myotubes and single isolated skeletal muscle fibres.

The main aim of the study is the development, optimisation, and validation of the methodological approaches to determine glucose uptake in the model of C2C12 myotubes since it is more feasible and avoids the use of animals. The second part of the study involves the application and adaptation of this methodology to a model of single skeletal muscle fibres isolated from flexor digitorum brevis (FDB) mouse muscle, which is more appropriate to study glucose uptake in pathological conditions associated with insulin resistance that appear in pathologies such as diabetes, obesity, and ageing. In parallel, we will investigate how RONS and redox homeostasis might affect the glucose uptake in skeletal muscle. As mentioned above, we hypothesised that RONS plays an important role as mediators in glucose uptake. The current knowledge in this area is scarce and limited. However, it is promising in the field of redox biology that the activity of RONS and redox homeostasis might be potential targets for the therapeutic intervention of insulin resistance in diabetes, obesity, and ageing [6,12].

## 2. Results

### 2.1. 6-NBDG/Glucose Uptake in C2C12 Myotubes: Insulin and Hydrogen Peroxide Effects

#### 2.1.1. Bovine Insulin: 6-NBDG Incubation 1 h

Myotubes were incubated with 6-NBDG for 1 h. As the negative control, we used myotubes without 6-NBDG, which showed fluorescence (autofluorescence) that was significantly lower compared from that of myotubes incubated with 6-NBDG (positive control). Myotubes treated with bovine insulin showed higher fluorescence compared to untreated myotubes (positive control). Myotubes exposed to H_2_O_2_ presented values of fluorescence that were slightly lower than those of the positive control myotubes, and the result was significant in the case of the 10 μM H_2_O_2_ group. When myotubes were concomitantly treated with bovine insulin and exposed to H_2_O_2_, fluorescence increased slightly compared to corresponding untreated myotubes. We observed that 6-NBDG provided fluorescence to myotubes, indicating that this glucose, 6-NBDG, was incorporated into the myotubes, and bovine insulin led to a slight increase in glucose uptake. Conversely, H_2_O_2_ might decreased glucose uptake under these experimental conditions (Figure 1).

#### 2.1.2. Recombinant Human Insulin

##### Recombinant Human Insulin: 6-NBDG Incubation 1 h

In this experimental assay, myotubes were incubated for 1 h in the presence of 6-NBDG. Myotubes treated with human recombinant insulin for 1 h presented values of fluorescence that were significantly higher compared from those of the control. Myotubes exposed to H_2_O_2_ for 2 h showed higher values of fluorescence that were significant in myotubes exposed to 10 μM H_2_O_2_. Myotubes treated with insulin and exposed to H_2_O_2_ concomitantly showed lower fluorescence compared from the myotubes treated with insulin alone, with significant results observed for myotubes treated with insulin and exposed to either 10 or 100 μM H_2_O_2_. Therefore, there was confirmation that human recombinant insulin produces glucose uptake in the model of myotubes and that 2 h of exposure to hydrogen peroxide also improves glucose uptake. However, there might be an interaction between insulin and hydrogen peroxide, since the expected increase in glucose uptake induced by insulin did not appear when myotubes were exposed concomitantly to H_2_O_2_ (Figure 2).

##### Recombinant Human Insulin: 6-NBDG Incubation Overnight

Unlike the previous assay, in this experiment, myotubes were incubated in the presence of 6-NBDG overnight. Myotubes treated with human recombinant insulin overnight presented values of 6-NBDG fluorescence significantly higher than those of the control. Myotubes exposed to H_2_O_2_ overnight showed similar fluorescence levels to those of the control myotubes. Myotubes treated with insulin and exposed to H_2_O_2_ concomitantly overnight showed lower fluorescence compared to those treated with insulin alone and higher fluorescence compared from the respective controls exposed to H_2_O_2_ 10 or 100 μM. Therefore, there was confirmation that recombinant human insulin yields 6-NBDG/glucose uptake in the model of myotubes and that long exposure to hydrogen peroxide does not affect glucose uptake. In addition, insulin was found to improve glucose uptake in myotubes exposed to hydrogen peroxide overnight (Figure 3).

### 2.2. 6-NBDG/Glucose Uptake in C2C12 Myotubes: Effects of Insulin Receptor Blocker and Angiotensin II—6-NBDG Incubation Overnight

When myotubes were preincubated with both peptide S-961, an antagonist for the insulin receptor, and human recombinant insulin, the fluorescence of the myotubes was significant lower compared to the fluorescence observed in myotubes treated with insulin alone. Myotubes treated with angiotensin II, 2.5 μM, showed lower fluorescence compared from the control, and when myotubes were cotreated with superoxide dismutase, the fluorescence was even lower. Myotubes treated with a higher concentration of angiotensin II, 10 μM, showed a dramatical increase in fluorescence, compared to the control. However, when myotubes were cotreated with superoxide dismutase, the fluorescence was similar to the fluorescence of control myotubes. The insulin receptor antagonist S-961 blocks the binding of insulin to its receptor and avoids the glucose uptake produced by insulin in myotubes. Angiotensin II 10 μM produced a dramatic increase of glucose uptake in myotubes; however, this effect was attenuated when myotubes were cotreated with superoxide dismutase (Figure 4).

### 2.3. 6-NBDG/Glucose Uptake in C2C12 Myotubes: Effect of Antioxidant Glutathione Depletion Induced by Diethyl Maleate and Diamide—6-NBDG Incubation Overnight

The treatment of myotubes with diethyl maleate (DEM), a glutathione oxidant, produced an increase of fluorescence compared from control myotubes, which was similar to the fluorescence of myotubes pre-treated with a glutathione donor, GSHEE, and diethyl maleate. Moreover, when myotubes were pre-treated with insulin, instead of GSHEE, the fluorescence appeared to be higher. Myotubes treated with diamide, which depletes intracellular GSH, showed a dramatical increase in fluorescence compared from the control myotubes. However, when myotubes were pre-treated with the glutathione donor GSHEE and diamide, fluorescence decreased, although the levels were still higher compared to those of the control. The pre-treatment of myotubes with insulin instead of GSHEE produced higher fluorescence in myotubes. The depletion of glutathione, the main intracellular antioxidant, appears to facilitate a prooxidant intracellular environment that favoured glucose uptake, which might be additive to the glucose uptake induced by insulin. When the oxidant environment became attenuated, at least partially due to the replenishment of GSH by GSHEE, the increase in glucose uptake was partially reduced. In addition, we observed that either DEM or diamide produced toxicity in myotubes, which was avoided when myotubes were pre-incubated with glutathione donor GSHEE. Insulin appeared to protect against toxicity induced by DEM but not against toxicity induced by diamide (Figure 5).

### 2.4. 6-NBDG/Glucose Uptake in C2C12 Myotubes: Effects of Cytochalasin B and Oxidants (Diethyl Maleate and Diamide)—6-NBDG Incubation Overnight

The treatment of myotubes with cytochalasin B, which is an inhibitor of glucose transport, did not produce changes in fluorescence. However, when myotubes were concomitantly treated with insulin and cytochalasin B, fluorescence of the myotubes decreased, although this decrease was not significant. Myotubes treated with diethyl maleate showed higher fluorescence compared to the control nontreated myotubes, although not statistically significant. The concomitant treatment with diethyl maleate and cytochalasin B produced a significant increase in fluorescence. The treatment with diamide produced a dramatically significant increase in fluorescence, and this effect remained when myotubes were cotreated with diamide and cytochalasin B. It appears that cytochalasin B might decrease the glucose uptake produced by insulin. In contrast, cytochalasin B might potentiate the glucose uptake induced by diethyl maleate and was not observed to affect the increase in glucose uptake produced by diamide (Figure 6).

### 2.5. 6-NBDG/Glucose Uptake in C2C12 Myotubes: Effect of Nitric Oxide—6-NBDG Incubation Overnight

Figure 7 shows myotubes that were exposed to three different nitric oxide donors: SNAP, NOC7, and SNP. In the case of exposure to SNAP, the fluorescence of myotubes increased significantly compared from the control (no exposure). Exposure to the nitric oxide donor NOC7 tended to reduce fluorescence slightly, although the result was not statistically significant. SNP (10 mM) produced toxicity and compromised the viability of the myotubes.

### 2.6. 6-NBDG/Glucose Uptake in Isolated Skeletal Muscle Fibres: Effects of Insulin and Hydrogen Peroxide—6-NBDG Incubation Overnight

Figure 8 presents fluorescence of skeletal muscle fibres isolated from 9- to 10-month-old male and female mice. The negative control is a group of fibres that were incubated in a medium without 6-NBDG; thus, the fluorescence of those fibres was autofluorescence. The other three groups included fibres incubated with 6-NBDG. Fibres exposed to 10 μM hydrogen peroxide presented higher values of fluorescence compared from both non-treated fibres (positive control) and fibres treated with insulin, 100 nM. These results revealed that hydrogen peroxide facilitated 6-NBDG/glucose uptake in isolated skeletal muscle fibres.

### 2.7. 6-NBDG/Glucose Uptake in Isolated Skeletal Muscle Fibres: Effects of Insulin, Hydrogen Peroxide, and Angiotensin II—6-NBDG Incubation Overnight

Figure 9 shows the 6-NBDG fluorescence of skeletal muscle fibres isolated from 19-month-old male and female mice. Although there is no statistical difference (ANOVA, *p* > 0.05), the fluorescence of fibres treated with insulin was higher than that of the control. Moreover, the fluorescence of fibres exposed to hydrogen peroxide and that of fibres treated with angiotensin II were higher than those of the control. Insulin, hydrogen peroxide, and angiotensin II appeared to favour glucose uptake in these fibres.

### 2.8. 6-NBDG/Glucose Uptake in Isolated Skeletal Muscle Fibres from Aged Mice: Effects of Insulin, Hydrogen Peroxide, and Nitric Oxide—6-NBDG Incubation Overnight

Figure 10 presents the 6-NBDG fluorescence of skeletal muscle fibres isolated from 25- to 29-month-old male mice. Assuming there are few fibres and no statistical significance (ANOVA, *p* > 0.05), the fluorescence of fibres treated with insulin is similar here to the fluorescence of the control fibres. However, the fluorescence of fibres exposed to hydrogen peroxide was higher than the fluorescence of the control fibres, considering there is not statistical difference. In the case of fibres treated with SNP, there was only one viable fibre, which presented an atypical morphology. Although it might be unreliable accounting just one fibre for any comparison, the fluorescence of this fibre was lower compared to the control group. Insulin was not found to facilitate glucose uptake in these fibres. However, hydrogen peroxide might slightly favour glucose uptake, assuming all statistical reservations.

## 3. Discussion

The main aim of this study was to develop a feasible methodology to determine glucose uptake in live skeletal muscle cells. This methodology is based on three pillars: (i) a glucose analogue, 6-NBDG, which is resistant to being metabolised and emits fluorescence; (ii) two models of live skeletal muscle cells, C2C12 myotubes and isolated mature fibres; and (iii) a quantitative fluorescence microscopy imaging analysis system used to determine 6-NBDG/glucose uptake by cells.

In the first instance, we assessed whether this methodology might detect glucose uptake in the model of C2C12 myotubes. These skeletal muscle cells were treated with insulin, the main physiological stimulus for glucose uptake. In addition, other myotube cultures were exposed to different concentrations of hydrogen peroxide. Several recent studies indicated that ROS may modulate and favour glucose uptake [9,10,11], and we had some experimental evidence in our laboratory, indicating that hydrogen peroxide, an important signalling ROS, might facilitate glucose uptake in C2C12 myotubes (data not published). Initially, we incubated myotubes with 20 μM 6-NBDG for 1 h; then, we removed the 6-NBDG and incubated the solution overnight in a Krebs solution buffer until the registration of fluorescence microscopy images. These myotubes emitted fluorescence that corresponded with autofluorescence (fluorescence emitted by intracellular molecules with fluorescence properties) and fluorescence originating from molecules of 6-NBDG that were internalised through GLUT4 receptors and retained in cytosol (Figure 1). It was reported that 6-NBDG is a fluorescence glucose analogue that is not metabolised [18,19]. Thus, every molecule that is transported into the cytosol remain in it and contribute to fluorescence emissions since 6-NBDG is not degraded or metabolised. We argue that this factor may be an advantage of 6-NBDG compared to other fluorescence glucose analogues such as 2-NBDG, which might be metabolised once internalised into the cytosol, and this activity might decrease the fluorescence emitted. Protocols based on 2-NBDG have been developed to determine glucose uptake in different cell types, such as astrocytes [20], endothelial cells [23], and skeletal muscle cells—both myotubes [14,24,25,26,27,28,29,30,31] and mature fibres isolated from FDB mouse muscle [15,16,17]. In these studies, cells were incubated with 2-NBDG for 15–60 min in a glucose free solution or medium, and the fluorescence was measured immediately. In some cases, this process was performed after harvest and the disruption of cells to measure the fluorescence of intracellular content, and in other studies, the fluorescence was measured from the whole well where cells grew. In every case, 2-NBDG is located in the cytosol of live cells for no more than 1 h, likely to avoid a loss of fluorescence since 2-NBDG is susceptible to be metabolised. Our methodological proposal is different, as we used 6-NBDG, which is resistant to metabolism and can maintain fluorescence in live cells for long period of time [18,19]. Thus, in our first set of experiments, myotubes were loaded with 6-NBDG for 1 h and then wash out. However, 6-NBDG remained overnight in the cytosol of live cells until fluorescence was registered by fluorescence microscopy imaging on specific areas from individual live myotubes. Figure 1 shows that myotubes loaded with 6-NBDG emitted fluorescence, which came from intracellular 6-NBDG that was transported into the cytosol alongside autofluorescence. This fluorescence, which was extrapolated as 6-NBDG/glucose uptake, was significantly higher compared from the fluorescence of myotubes that were not incubated with 6-NBDG but emitted autofluorescence.

Insulin is the main physiological stimulus for glucose uptake in skeletal muscle cells. We assessed this phenomenon in our model incubating myotubes with 100 nM insulin (bovine origin) for 2 h and proved that 6-NBDG uptake increased compared to the control myotubes (Figure 1). However, this increase was statistically not significant. This made us suspect that insulin might have partially degraded, impairing its function. Thus, in another experiment set, we changed to recombinant human insulin 100 nM with incubation for 2 h. In this experiment, myotubes showed a statistically significant increase in 6-NBDG fluorescence uptake, which was assumed to be glucose uptake (Figure 2). This result demonstrated that our model was able to assess the glucose uptake evoked by insulin in myotubes, and we decided to use recombinant human insulin for subsequent experiments. There are different factors, such as pharmacological, functional (exercise), and nutritional interventions, among others, that may potentially affect glucose uptake. However, the modulation of glucose uptake induced by those factors is observed over time rather than to be an acute effect. This rationale supports that the experimental design of studies to investigate the effects produced by factors such as reagents, drugs, functional and nutritional interventions, ageing, etc. in glucose uptake and insulin resistance in skeletal muscle must be planned as a long-time course experiment. Therefore, we designed the following experiment set to assess glucose uptake in myotubes incubated with insulin and/or H_2_O_2_ at different concentrations in the presence of 6-NBDG for at least 19 h (overnight). In this experimental set (Figure 3), we observed that the fluorescence of myotubes was similar compared to that in previous experiments (Figure 1 and Figure 2), where 6-NBDG was present for 1 h and insulin and H_2_O_2_ incubation were performed for 2 h. Moreover, we verified the effects of insulin, which produced a significant increase in glucose uptake compared from the control. This result demonstrated the suitability of 6-NBDG, which remained in the cytosol without fluorescence attenuation, since this molecule is resistant to degradation and metabolisation [18,19] and provides a fluorescence signal to monitor glucose uptake in long-term experiments. These results established the fundamentals for the subsequent long-term experiments, where we studied the effect of RONS and reagents that affect intracellular redox homeostasis of glucose uptake in myotubes and skeletal muscle fibres.

It was described that ROS and nitric oxide may facilitate glucose uptake in skeletal muscle [9,10,11]. Firstly, we assayed whether hydrogen peroxide, a signalling ROS [32], affects glucose uptake in myotubes and carried out experiments where myotubes were exposed to different concentrations of H_2_O_2_: 1, 10, and 100 μM (Figure 1, Figure 2 and Figure 3). We observed that myotubes exposed for 20 h to H_2_O_2_ 10 and 100 μM presented similar glucose uptake to the control myotubes and found that glucose uptake improved when myotubes were incubated for 20 h with H_2_O_2_ and insulin concomitantly (Figure 3). Therefore, in these experiments, the effect of H_2_O_2_ as a potential inductor of glucose uptake was unclear. Moreover, it appears that there might be an interaction between insulin and hydrogen peroxide, since when hydrogen peroxide and insulin are concomitantly present for two hours, the glucose uptake increase evoked by insulin is attenuated (Figure 2). We speculate, that this kind of negative interference might be originated by the oxidant properties of hydrogen peroxide. Since this ROS, hydrogen peroxide, might be producing oxidative damage in the insulin molecule, its receptor, or in downstream elements of the PI3K/Akt signalling pathway. This might explain the decrease of expected fluorescence/glucose uptake evoked by insulin. However, this hypothesis must be addressed in future investigations that currently are out of the scope in the present study.

In another experimental set, we proved that the blocking of the insulin receptor with peptide S-961, an antagonist for insulin receptors [33], avoided glucose uptake induced by insulin in myotubes (Figure 4). This result may indicate that when insulin binding to its receptor is impaired, there is a reduction in glucose uptake. There could be interference with the downstream PI3K/Akt signalling pathway that modulates the translocation of GLUT4 to the plasma membrane and leads to glucose uptake [6,34].

It has been reported that angiotensin II, through binding to its receptor AT1 and following the activation of NADPH oxidase, stimulates the production of ROS, which contributes to the development of insulin resistance in skeletal muscle [35,36,37]. In our experiments, 10 μM angiotensin II induced a significant increase in glucose uptake in C2C12 myotubes (Figure 4). This increase in glucose uptake conflicted with our expected results: impairment of glucose uptake and insulin resistance produced by angiotensin II. This result might be due to the high amount of ROS generated by angiotensin II and NADPH oxidase activation interfering with the insulin PI3K/Akt signalling pathway that facilitates glucose uptake [35,36,37]. However, in our experiment, there was no insulin, so this pathway might have been unaffected. Instead, the amount of ROS generated via the activation of NADPH oxidase coupled to the AT1 receptor when angiotensin II binds to it [37,38] might be moderated, and this might facilitate glucose uptake through the AMPK signalling pathway, which is independent of insulin [11,39]. This hypothesis may be confirmed in future experiments that currently are beyond the scope of the present study.

In other experiments, myotubes treated with angiotensin II were extracellularly exposed to superoxide dismutase (Figure 4). The aim was to transform the potential superoxide generated extracellularly by NOX2 due to angiotensin II activation [10,39,40,41] into H_2_O_2_, which might diffuse into the cytosol, thereby increasing the intracellular concentration of H_2_O_2_. The results revealed that 10 μM angiotensin II and SOD exposure avoided the increase in glucose uptake produced by angiotensin II. Likely, the superoxide extracellularly produced by NOX2 was converted into H_2_O_2_ by SOD and then diffused into the cytosol, contributing to an increase in the pool of H_2_O_2_ generated intracellularly by intracellular SOD. Thus, the intracellular concentration of H_2_O_2_ is high, and this produced oxidative distress [32], which might block AMPK signalling and avoid, at least in part, glucose uptake. Again, this hypothesis may be assessed in future complementary studies, which are currently outside the scope of this study. In addition, isolated skeletal muscle fibres from 19-month-old mice treated with 100 nM angiotensin II and free from insulin exposure showed higher, albeit not statistically significant, glucose uptake compared to non-treated control fibres (Figure 9). This result indicated that angiotensin II somehow facilitates glucose uptake in skeletal muscle fibres.

It was previously proposed that intracellular RONS may facilitate glucose uptake in skeletal muscle either through activation of the AMPK signalling pathway [11,39] or independently from this pathway [40]. We investigated whether this effect was recapitulated in our model. For this analysis, myotubes were treated with diethyl-maleate (DEM), a strong oxidant that depletes the main intracellular antioxidant defence, glutathione (GSH), and might increase intracellular ROS and yield oxidative stress [41,42]. Myotubes treated with DEM showed a higher increase of glucose uptake compared from the control, at the limit of statistical significance (Figure 5). Therefore, it can be speculated that the intracellular pro-oxidation redox state induced by DEM drives the increase in glucose uptake that was observed in myotubes. When myotubes were treated concomitantly with DEM and GSHEE, a GSH donor [43,44,45], the glucose uptake was similar to that in myotubes treated with DEM. Thus, it appears that GSH supplied to myotubes did not revert the oxidation produced by DEM, and this intracellular level of oxidation facilitated glucose uptake. The concomitant treatment of DEM and insulin produced an additional increase in glucose uptake. This result could be explained by the insulin activation of the PI3K/Akt signalling pathway, which facilitates glucose uptake [6,46,47]. In addition, we analysed the effects of another strong oxidant, diamide, that was reported to produce oxidative stress in cells and, consequently, lead to high levels of ROS [48,49]. Thus, myotubes treated with diamide experienced a significant increase in glucose uptake, which was avoided when the myotubes had been cotreated with GSHEE and consequently GSH had been replenished (Figure 5). This result indicates that high intracellular oxidation might facilitate glucose uptake due to elevated levels of ROS. However, the intracellular reductant environment, due to GSH replenishment and the subsequent decrease in ROS level, can partially avoid the increase in glucose uptake produced by ROS. When myotubes were cotreated with diamide and insulin, we observed an additive effect in glucose uptake (Figure 5). This effect could indicate that the induced-insulin PI3K/Akt signalling pathway was also activated, helping to improve glucose uptake.

We investigated whether our methodology for monitoring 6-NBDG uptake may be suitable to detect the inhibition of glucose uptake produced by the interruption of glucose transport using cytochalasin B, which is a blocker of GLUT transporters [18,31]. The impairment of glucose transport induced by cytochalasin B has been used in different cellular models, including skeletal muscle cells, to determine basal glucose uptake using radioactive 2-deoxyglucose [50,51,52]. In a study using the fluorescence glucose analogue 2-NBDG, Osorio-Fuentealba et al. reported that cytochalasin B reduces glucose uptake in skeletal muscle cells [31]. However, in our experiments, it was difficult to observe whether cytochalasin B impaired 6-NBDG uptake in myotubes (Figure 6). It appeared that glucose uptake was slightly reduced in myotubes treated with insulin and cytochalasin B, along with an unexpected increase of 6-NBDG uptake in those myotubes treated with DEM and cytochalasin B. Moreover, cytochalasin B did not avoid the 6-NBDG uptake produced by diamide. Therefore, cytochalasin B did not seem to affect glucose transport in our experiments. This result conflicts with reports indicating that cytochalasin B impairs glucose uptake [31,50,51,52]. Meanwhile, we observed that cytochalasin B affected the viability of myotubes (images in Figure 6). This toxicity induced by cytochalasin B might be due to the partial blocking of GLUT4 since cytochalasin B is an inhibitor of the GLUT4 transporter [21,22], which might reduce the internalisation of glucose and thus compromise the intracellular glucose supply, possibly leading to the impairment of cellular metabolism and viability. However, it appeared that intracellular 6-NBDG fluorescence was unchanged. This result could be explained in the following way. Although glucose uptake was impaired by cytochalasin B, it was reported that 6-NBDG binds to GLUT1 transporter anchored in the plasma membrane of astrocytes with 300 times higher affinity than glucose [19], and GLUT1 receptors are abundant in the membrane of C2C12 myotubes [53]. In the present study, the myotubes were exposed overnight to 6-NBDG and glucose in the culture medium, and the high affinity of 6-NBDG to bind to GLUT transporters might explain the internalised molecules of 6-NBDG, which have high affinity for GLUT receptors, and once those molecules are in the cytosol un-metabolised, they emit fluorescence. This factor could complicate monitoring the impairment of glucose uptake produced by cytochalasin B. This result could indicate that 6-NBDG improperly monitors the impairment of glucose uptake produced by cytochalasin B. However, 6-NBDG is very useful to monitor the improvements to glucose uptake produced by insulin, ROS, and the oxidant environment. These factors facilitate the translocation of GLUT4 receptors from the cytosol to the plasma membrane and increase the population of GLUT4 in the plasma membrane, driving the increase of glucose and 6-NBDG internalisation. Thus, it is possible to monitor the improvement of glucose uptake. 6-NBDG is suitable to measure increases in glucose uptake (accumulative 6-NBDG fluorescence), but proper reservations must be considered when a reduction in basal glucose uptake is under analysis.

It was reported that, in addition to ROS, nitric oxide might stimulate the translocation of GLUT4 to the plasma membrane and facilitate glucose uptake in skeletal muscle [11]. We sought to probe this hypothesis in our model of myotubes. Thus, we carried out an experiment where myotubes were incubated in the presence of nitric oxide donors (Figure 7). Incubation in the presence of the nitric oxide donor SNAP produced a significant increase in 6-NBDG uptake in myotubes. It was demonstrated in single skeletal muscle fibres that the decomposition of SNAP extracellularly releases nitric oxide, which might diffuse through the plasma membrane into the cytosol, increasing the intracellular concentration of nitric oxide [54]. It is plausible that, in our model, the intracellular augmentation of nitric oxide in myotubes could drive the translocation of GLUT4 to the plasma membrane, which might facilitate 6-NBDG and glucose uptake. It is also possible that this process would be mediated by AMPK signalling activation, as reported in a study carried out by Higaki et al., which demonstrated in the model of isolated soleus skeletal muscle that NO stimulated glucose uptake through a different mechanism than the insulin and contraction signalling pathways [55]. However, there is a need for further investigations to uncover the mechanism of nitric oxide-induced glucose uptake in skeletal muscle. In addition, we used another nitric oxide donor, NOC-7, and observed no increase in 6-NBDG uptake. This result was unexpected compared to the increase in glucose uptake produced by SNAP. Therefore, we speculate that the NOC-7 we used for the experiments might have been out of date, or the concentration was too low to release enough NO. In any case, we observed insufficient intracellular NO to evoke glucose uptake. To confirm this hypothesis, in future studies, we will perform additional experiments using new biosensors for NO intracellular detection, a methodology that we recently developed and optimised [56] to properly assess the intracellular concentration of NO. In the case of the nitric oxide donor SNP, we observed toxicity and the absence of myotube to assess 6-NBDG uptake. For future experiments, we proposed using a lower concentration of SNP to avoid toxicity. As mentioned before, we recently developed and optimised a new methodology based on the use of genetically encoded biosensors to detect and monitor, in real time, the intracellular concentration of NO in live cells [56]. Therefore, we propose to use this methodology to determine the intracellular NO concentration and uncover whether this concentration facilitates 6-NBDG and glucose uptake in myotubes and fibres.

The research undertaken with the model of C2C12 myotubes was at the core of the development and adaptation of this methodology based on 6-NBDG to determine and quantify glucose uptake in the model of single isolated skeletal muscle fibres. For fibres, we incorporated a new adaptation for the calculation of 6-NBDG uptake by fibres. This calculation was taken as the fibre net fluorescence, calculated from the fluorescence emitted by the fibres and subtracting the extracellular fluorescence, which is the fluorescence of the medium and equivalent to the background. Although we first tried this approach in experiments with myotubes, we realised that it was inadequate since the density of cells (myoblasts and myotubes) growing in the culture was close to 100% and, in practise, it is very difficult to find areas without cells in microscopy field images that could be selected to quantify the fluorescence of the medium. In the case of fibres, it is always possible to find an extracellular area close to the fibre that is free of cells and could be selected to quantify the fluorescence of the medium (background).

The first set of experiments was performed with fibres isolated from 9–10-month-old mice (Figure 8). Fibres that were incubated without 6-NBDG, i.e., the negative control, emitted autofluorescence. As in myotubes, this autofluorescence is intrinsic and produced by intracellular components with fluorescence properties. Fibres incubated with 6-NBDG, i.e., the positive control, presented values of fluorescence higher than those of the negative control, with no statistical difference compared to the fibres treated with insulin. It is well known that insulin is the main factor that facilitates glucose uptake under physiological conditions [6], and we expected an increase in 6-NBDG. However, we observed no change in 6-NBDG fluorescence in fibres treated with insulin. Thus, we speculated that there might be an impairment in the induced-insulin PI3K/Akt pathway that regulates glucose uptake or a blockade in insulin binding to its receptor. Somehow, these fibres appeared to show insulin resistance. However, the fibres were isolated from young-adult mice 9–10 months old; thus, insulin resistance due to ageing is unlikely. Other reasons could include external experimental factors out of our control (pH, temperature, and CO_2_) that affect glucose uptake. The fibres treated with the ROS hydrogen peroxide yielded a significant increase in 6-NBDG, which confirmed that hydrogen peroxide and a moderate oxidative state facilitated glucose uptake in skeletal muscle fibres, although these fibres might be insulin resistant. This result reinforces the hypothesis that oxidative eustress produced by hydrogen peroxide [57] facilitates glucose uptake in skeletal muscle [58]. This is our hypothesis, which we believe will yield functional and pharmacological therapeutic approaches in the future if this research continues.

We performed a set of experiments with fibres isolated from 19-month-old mice (Figure 9). Fibres treated with insulin, hydrogen peroxide, or angiotensin II showed an increase in 6-NBDG uptake compared to the control. Although this increase was not significant, the tendency showed a clear increase in 6-NBDG uptake. These results support the notion that insulin favours glucose uptake, but this finding might be slightly impaired, likely due to the fact that fibres were isolated from adult-old mice, and ageing is one of the main factors for the development of insulin resistance [5]. In addition, hydrogen peroxide might facilitate glucose uptake [58], even if the fibres are insulin resistant due to ageing. Moreover, a prooxidative state induced by angiotensin II that activates NADPH oxidases, which are abundantly expressed in skeletal muscle fibres [59], would drive the generation of superoxide and hydrogen peroxide, which might facilitate glucose uptake in these fibres from adult-old mice, who could be insulin resistant.

The last set of experiments examined skeletal muscle fibres isolated from very old mice, 25–29 months old. It was observed that 6-NBDG uptake in fibres treated with insulin was similar to that in the control fibres without treatment, whereas hydrogen peroxide appeared to slightly improve 6-NBDG uptake. In the case of the nitric oxide donor SNP, there was only one fibre that showed an atypical morphology while other remaining fibres appeared damaged, with signs of toxicity or damage. This might be due to the increase in the intracellular concentration of nitric oxide in these fibres making such fibres prone to the reaction of nitric oxide with superoxide to generate peroxynitrite, which is a very reactive RONS that produces deleterious oxidative damage in proteins and lipids and might produce the toxicity and damage observed in fibres [60]. It appears that these fibres from very old mice were insulin resistant, with insulin unable to revert glucose uptake impairment, whereas hydrogen peroxide might slightly facilitate glucose uptake. These results are preliminary, and there is still a need for further investigations to uncover whether the induced moderate oxidative homeostasis might facilitate glucose uptake in skeletal muscle in ageing.

## 4. Materials and Methods

### 4.1. Reagents

The hydrogen peroxide solution (30 wt% in water), S-nitroso-N-acetylpenicillamine (SNAP), Sodium nitroprusside dihydrate (SNP), NOC-7, Diamide, Diethyl Maleate (DEM), Glutathione reduced ethyl ester (GSHEE), Superoxide dismutase (SOD), Angiotensin II, Cytochalasin B, bovine insulin, and recombinant human insulin were obtained from Merck-Sigma-Aldrich (Darmstadt, Germany). Insulin receptor antagonist peptide (S961) was obtained from Phoenix Pharmaceuticals, Inc. (Burlingame, CA, USA). 6-(N-(7-Nitrobenz-2-oxa-1,3-diazol-4-yl) amino)-6-Deoxyglucose (6-NBDG) was obtained from Molecular Probes^TM^, Invitrogen, Thermo Fisher Scientific (Waltham, MA, USA).

### 4.2. Skeletal Muscle Cell Culture

The C2C12 mouse skeletal myoblasts were purchased from the American Type Culture Collection (CRL-1772, Manassas, VA, USA). C2C12 was cultured in Dulbecco’s modified Eagle’s medium (DMEM; Sigma-Aldrich) containing 10% (*v*/*v*) foetal bovine serum (FBS) (Invitrogen, Waltham, MA, USA), 2 mM L-glutamine (Sigma-Aldrich), 50 i.u. penicillin, and 50 µg of mL-1 streptomycin (Sigma-Aldrich) at 37 °C in an atmosphere of 5% CO_2_ under adequate humidity. C2C12 myoblasts were differentiated for 7 days to obtain myotubules; during this time, the myoblasts fused and transformed into multinuclear myotubes with a tubular morphology. Differentiation involved growing myoblasts to 80%–90% confluency and then replacing the growth medium with a differentiation medium (DMEM with 2% (*v*/*v*) horse serum (Invitrogen) supplemented with 2 mM L-glutamine, 50 i.u. penicillin, and 50 µg mL-1 of streptomycin) [61].

### 4.3. Animals

Female and male 9–29-month-old C57BL/6J mice were used in this study. The procedures involving animals were approved by the Bioethics Committee of the University of Salamanca and conducted in accordance with the Spanish (RD 53/2013) and European Union (2010/63/EU) guidelines for animal experimentation.

### 4.4. Skeletal Muscle Fibres Isolation

Single skeletal muscle fibres were isolated from the FDB muscle of the mice according to the protocol established by Palomero et al. [43,44,54,56,61]. FDB muscles were placed into 0.4% type I collagenase (EC 3.4.24.3; Sigma-Aldrich) solution in the culture medium (minimum essential medium (MEM; Sigma-Aldrich) supplemented with 10% foetal bovine serum (FBS; Invitrogen) containing 2 mM glutamine, 50 IU penicillin, and 50 μg/mL streptomycin. The mixture compose of FDB muscles and collagenase solution was incubated at 37 °C for 2 h and manually shaken every 30 min to improve digestion of connective tissue and the release of fibres. Free single muscle fibres were separated from broken fibres and debris by centrifuging at low speed (600× *g* for 30 s) four times. After each centrifugation, the supernatant was removed and replaced with fresh culture medium. Washed fibres were seeded onto a 13 mm diameter glass surface inserted in individual 35 mm culture dishes (MatTek Corporation, Ashland, MA, USA). The glass surface was previously coated with a collagen matrix (Matrigel, BD Biosciences, Franklin Lakes, NJ, USA). Finally, fibres were incubated for 18–24 h covered with culture medium at 37 °C in 5% CO_2_ in a humidified atmosphere to allow adherence of the fibres to the collagen matrix. Fluorescence microscopy experiments were undertaken only on selected fibres that displayed excellent morphology and exhibited clear striations along the sarcolemma.

### 4.5. Experimental Procedure

C2C12 myotubes and skeletal muscle fibres were exposed to a similar procedure. First, the C2C12 cell medium was changed to a Krebs solution (119 mM NaCl, 2.5 mM KCl, 2.5 mM CaCl_2_-2H_2_O, 1.5 mM MgSO_4_-7H_2_O, 1.25 mM NaH_2_PO_4_, 26.2 mM NaHCO_3_, and 11.1 mM glucose; pH 7.4). After one hour of stabilisation, we added the different reagents for the experimental treatments: bovine and recombinant human insulin, SNAP, SNP, Diamide, GSHEE, SOD, Angiotensin II, NOC-7, S961, DEM, Cytochalasin B, and hydrogen peroxide. After one hour, 6-NBDG was added to the cell medium and incubated with the reagents of treatments for another hour. After this time, in some experiments (Figure 1 and Figure 2), cells were washed with Krebs solution and maintained in a 5% CO_2_ humidified atmosphere at 37 °C overnight. In the rest of the experiments, myotubes were incubated with the reagents for treatment and 6-NBDG overnight (see the experimental procedure in the corresponding figures). In the case of skeletal muscle fibres, the complete procedure was undertaken with fibres incubated overnight in Minimum Essential Medium Eagle (MEM (M8042, Sigma-Aldrich^®^) with 10% (*v*/*v*) foetal bovine serum (Invitrogen) supplemented with 2 mM L-glutamine, 50 i.u. penicillin, and 50 µg mL^−1^ of streptomycin (Sigma-Aldrich)). After overnight incubation with the reagents for treatments and 6-NBDG, both myotubes and fibres in the culture plates were placed under the fluorescence microscope stage for the registration of images.

### 4.6. Fluorescence Microscopy and Quantitative Image Analysis

C2C12 myotubes and single skeletal muscle fibres were placed under a fluorescence microscope (Live Cell Observer, Carl Zeiss, Oberkochen, Germany) equipped with a chamber to maintain the temperature at 37 °C in a 5% CO_2_ atmosphere to register images of these cells. Images were obtained over a time-lapse that consisted in collecting images every 1 min for 5 min under 20× objective magnification. The source of excitation light was a light-emitting diode (LED) that generated 470 nm monochromatic light. Fluorescence images were obtained through a fluorescence cube with a 450/40 nm excitation filter, 495 nm beam splitter, and 525/50 nm emission filter. Fluorescence images were acquired with a computer-controlled CCD camera (AxioCam MRm, Carl Zeiss, Oberkochen, Germany) coupled to the microscope. The fluorescence images were acquired with the minimum exposure time that provided images with enough quality for quantification to avoid light-induced damage to the cells. Time-lapse images were acquired with the same exposure time, which was maintained for each time-lapse experiment performed under the same conditions. The fluorescence image analysis was undertaken using the software (ZEN 2 blue edition, Carl Zeiss) included with the microscopy equipment. This software provided the fluorescence quantification of myotubes and fibres from fluorescence images that had been recorded during the time-lapse experiment. For quantification, a region of interest (ROI) from a myotube or fibre that emitted fluorescence was selected. The software provided the measurement, in terms of grey activated pixels, of the fluorescence intensity from that ROI in every image recorded. The fluorescence intensity from a unique ROI was calculated as the mean of the 6 sequential images registered from the lime-lapse of 5 min, and this was considered as the fluorescence of that particular myotube or fibre. In the case of experiments with myotubes, one time lapse was performed per myotube culture dish, and a maximum of 5 individual myotubes in the image field were selected for quantification. In the case of fibres, one time-lapse per fibre culture dish was performed, and 1–3 viable fibres in the image field were selected for fluorescence quantification. Moreover, a ROI from the extracellular space of fibres was selected to measure fluorescence, and this fluorescence (background) was subtracted from the fluorescence of the fibre to obtain the net fluorescence of the fibre, which was then used for the subsequent analysis.

In addition, the light transmission images registered were used for the assessment of toxicity and viability, which might have been affected by the experimental interventions, in either myotubes or fibres. This assessment was based in the observation of morphology, density, and general aspect of myotubes and fibres.

### 4.7. Statistical Analysis

The values for the raw fluorescence of myotubes and net fluorescence of fibres, are presented as the mean ± standard error of the mean (SEM) and the number (n) of myotubes or fibres used for statistical analysis. Data were analysed with a one-way ANOVA to detect differences between experimental treatment groups, followed by a post hoc LSD test (Student’s *t*-test) to detect differences between the two experimental treatments. The statistical difference was set at *p* < 0.05. The IBM SPSS Statistics software, version 28.0.1.1 (14), was used for statistical analysis.

## 5. Conclusions

In this study, we adapted, developed, and validated a methodology based on the fluorescence glucose analogue 6-NBDG, combined with quantitative fluorescence microscopy image analysis to determine the glucose uptake in two models of skeletal muscle cells: C2C12 myotubes and single fibres isolated from skeletal muscle. The glucose uptake was determined specifically in live cells, observed under the microscope, which were maintained in a physiological medium without depriving any component (i.e., glucose) that might interfere with normal metabolism. In addition, glucose uptake was determined over a long period (20 h), which helped explore the effects of all treatments on the improvement of glucose uptake.

The method of 6-NBDG was suitable to determine an increase in glucose uptake greater than the basal glucose uptake. However, this methodology overestimated glucose uptake when basal glucose uptake was impaired (e.g., due to the blocking of glucose transport with cytochalasin B).

The prooxidative intracellular redox environment produced by RONS such as hydrogen peroxide and nitric oxide, which might be associated with oxidative eustress, improved glucose uptake in myotubes and skeletal muscle fibres. However, when oxidation was excessive (i.e., oxidative distress), cells could become damaged or experience toxicity, compromising cellular viability. Moreover, in this circumstance, glucose uptake appeared to be elevated.

In this article, we presented a new methodologic approach to determine 6-NBDG/glucose uptake in skeletal muscle cells. This methodology is feasible and valid and could be complemented with other research techniques, e.g., molecular biology and immunocytochemistry, to help uncover the mechanisms that produce a pathological state of skeletal muscle, insulin resistance, and impairment of glucose uptake, which are manifested in pathologies such as type 2 diabetes, metabolic syndrome, and obesity, as well as in ageing.

## Figures and Tables

**Figure 1 ijms-24-08082-f001:**
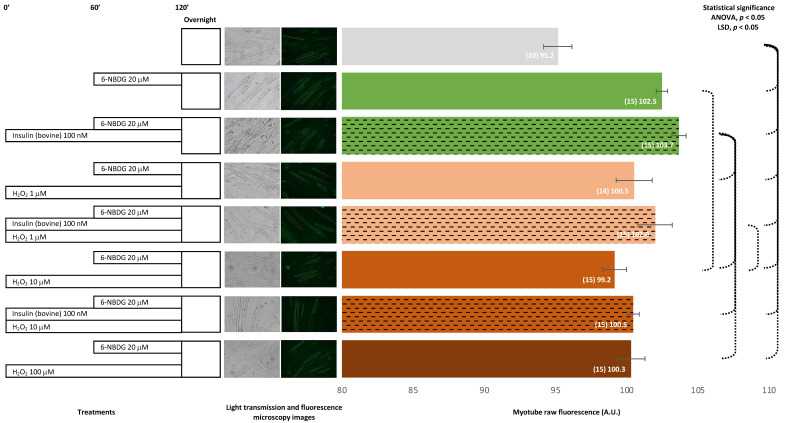
6-NBDG uptake in C2C12 myotubes under different treatments. 6-NBDG incubation for 1 h. Treatments exposure was 2 h: insulin (bovine origin), hydrogen peroxide, and insulin (bovine origin) and hydrogen peroxide. Data from individual myotubes cultured in three independent wells are presented as the mean of raw 6-NBDG intracellular fluorescence, number of myotubes (n), and SEM. Statistical analysis: one-way ANOVA, followed by the LSD post-hoc test for paired comparisons. Statistical significance was set to *p* < 0.05: ANOVA and LSD post-hoc (dashed connectors). Light transmission and fluorescence emission microscopy images show individual C2C12 myotubes representative of every experimental condition.

**Figure 2 ijms-24-08082-f002:**
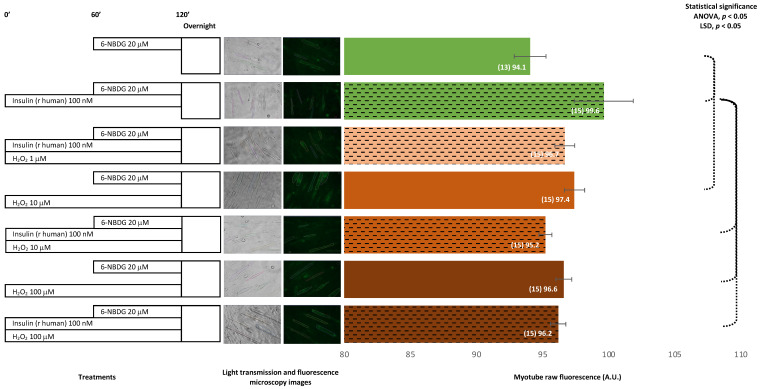
6-NBDG uptake in C2C12 myotubes under different treatments. 6-NBDG incubation for 1 h. Treatments exposure was 2 h: insulin, hydrogen peroxide, and insulin and hydrogen peroxide. Data from individual myotubes cultured in three independent wells are presented as the mean of raw 6-NBDG intracellular fluorescence, number of myotubes (n), and SEM. Statistical analysis: one-way ANOVA, followed by the LSD post-hoc test for paired comparisons. Statistical significance was set to *p* < 0.05: ANOVA and LSD post-hoc (dashed connectors). Light transmission and fluorescence emission microscopy images show individual C2C12 myotubes representative of every experimental condition.

**Figure 3 ijms-24-08082-f003:**
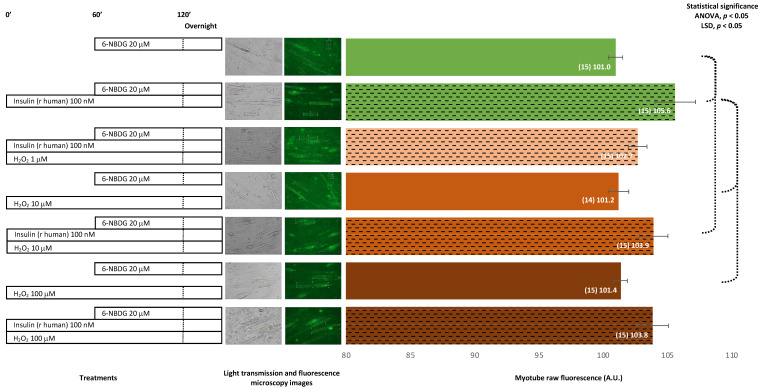
6-NBDG uptake in C2C12 myotubes under different treatments. 6-NBDG incubation for 19 h. Treatments exposure was 20 h: insulin, hydrogen peroxide, insulin, and hydrogen peroxide. Data from individual myotubes cultured in three independent wells are presented as the means of raw 6-NBDG intracellular fluorescence, number of myotubes (n), and SEM. Statistical analysis: one-way ANOVA, followed by the LSD post-hoc test for paired comparisons. Statistical significance was set to *p* < 0.05: ANOVA and LSD post-hoc (dashed connectors). Light transmission and fluorescence emission microscopy images show individual C2C12 myotubes representative of every experimental condition.

**Figure 4 ijms-24-08082-f004:**
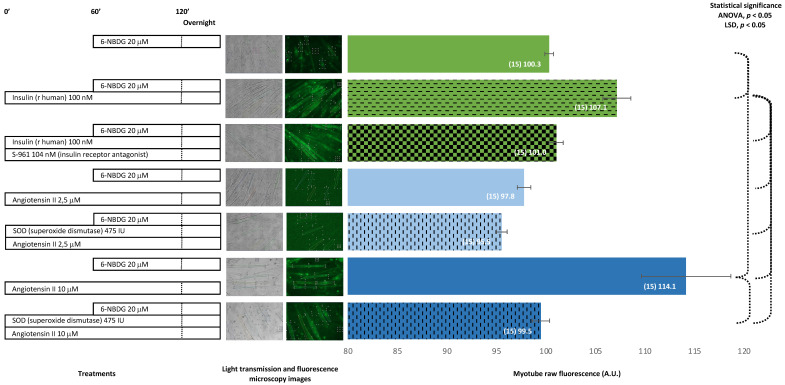
6-NBDG uptake in C2C12 myotubes under different treatments. 6-NBDG incubation for 19 h. Treatments exposure was 20 h: insulin, insulin receptor blocker (S-961) and insulin, angiotensin II, and superoxide dismutase and angiotensin II. Data from individual myotubes cultured in three independent wells are presented as the means of raw 6-NBDG intracellular fluorescence, number of myotubes (n), and SEM. Statistical analysis: one-way ANOVA, followed by the LSD post-hoc test for paired comparisons. Statistical significance was set to *p* < 0.05: ANOVA and LSD post-hoc (dashed connectors). Light transmission and fluorescence emission microscopy images show individual C2C12 myotubes representative of every experimental condition.

**Figure 5 ijms-24-08082-f005:**
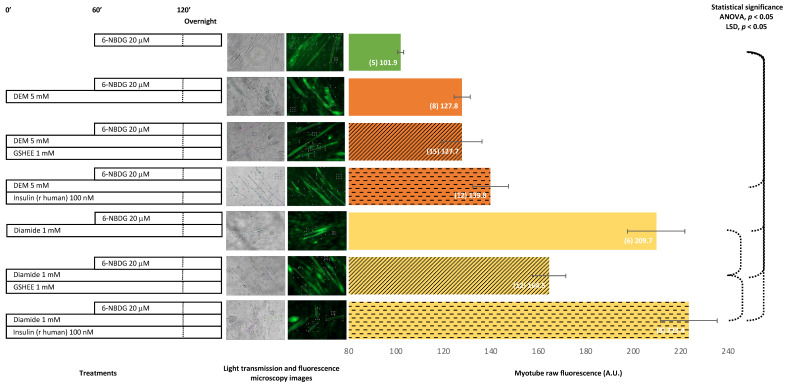
6-NBDG uptake in C2C12 myotubes under different treatments. 6-NBDG incubation for 19 h. Treatments exposure was 20 h: DEM, DEM and GSHEE, DEM and insulin, diamide, diamide and GSHEE, and diamide and insulin. Data from individual myotubes cultured in three independent wells are presented as the means of raw 6-NBDG intracellular fluorescence, number of myotubes (n), and SEM. Statistical analysis: one-way ANOVA, followed by the LSD post-hoc test for paired comparisons. Statistical significance was set to *p* < 0.05: ANOVA and LSD post-hoc (dashed connectors). Light transmission and fluorescence emission microscopy images show individual C2C12 myotubes representative of every experimental condition.

**Figure 6 ijms-24-08082-f006:**
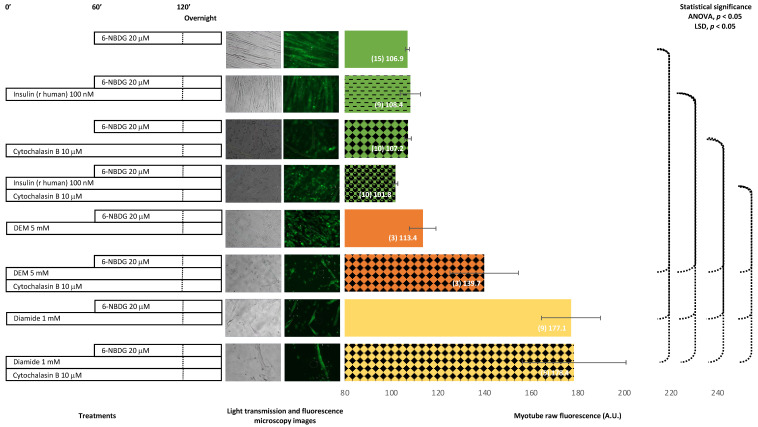
6-NBDG uptake in C2C12 myotubes under different treatments. 6-NBDG incubation for 19 h. Treatments exposure was 20 h: insulin, cytochalasin B, cytochalasin B and insulin, DEM, DEM and cytochalasin B, diamide, and diamide and cytochalasin B. Data from individual myotubes cultured in three independent wells are presented as the means of raw 6-NBDG intracellular fluorescence, number of myotubes (n), and SEM. Statistical analysis: one-way ANOVA, followed by the LSD post-hoc test for paired comparisons. Statistical significance was set to *p* < 0.05: ANOVA and LSD post-hoc (dashed connectors). Light transmission and fluorescence emission microscopy images show individual C2C12 myotubes representative of every experimental condition.

**Figure 7 ijms-24-08082-f007:**
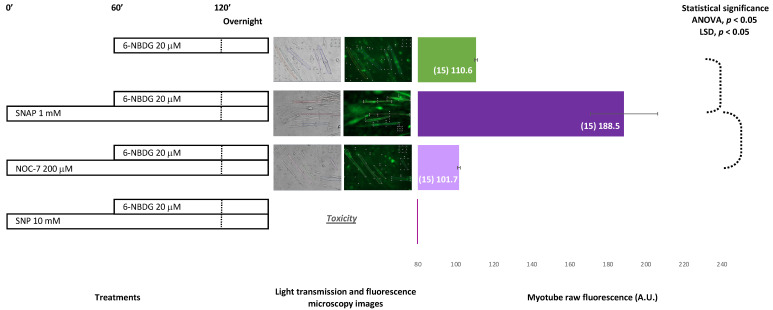
6-NBDG uptake in C2C12 myotubes under different treatments. 6-NBDG incubation for 19 h. Treatments exposure was 20 h: SNAP, NOC-7, and SNP. Data from individual myotubes cultured in three independent wells are presented as the means of raw 6-NBDG intracellular fluorescence, number of myotubes (n), and SEM. Statistical analysis: one-way ANOVA, followed by the LSD post-hoc test for paired comparisons. Statistical significance was set to *p* < 0.05: ANOVA and LSD post-hoc (dashed connectors). Light transmission and fluorescence emission microscopy images show individual C2C12 myotubes representative of every experimental condition.

**Figure 8 ijms-24-08082-f008:**
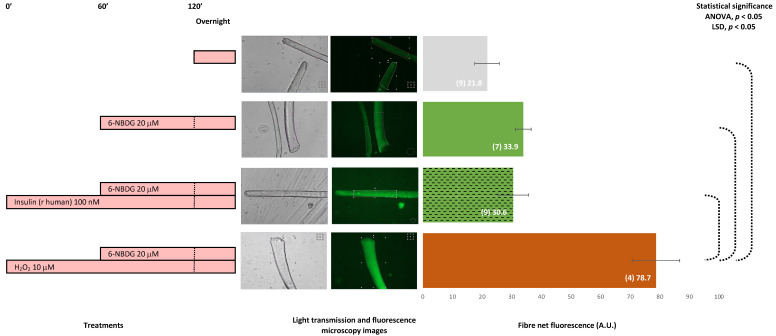
6-NBDG uptake in skeletal muscle fibres isolated from 9–10-month-old male and female mice. Fibres in the culture were placed under different treatments. 6-NBDG incubation for 19 h. Treatments exposure was 20 h: insulin and hydrogen peroxide. Data from individual fibres cultured in independent plates are presented as the means of net 6-NBDG intracellular fluorescence, number of fibres (n), and SEM. Statistical analysis: one-way ANOVA followed by LSD post-hoc test for paired comparisons. Statistical significance was set to *p* < 0.05: ANOVA and LSD post-hoc (dashed connectors). Light transmission and fluorescence emission microscopy images show individual fibres representative of every experimental condition.

**Figure 9 ijms-24-08082-f009:**
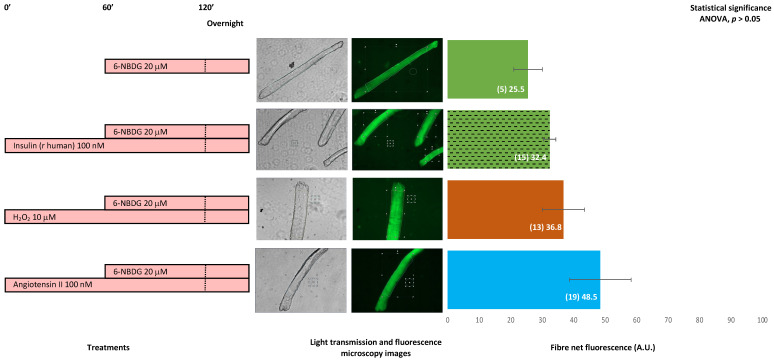
6-NBDG uptake in skeletal muscle fibres isolated from 19-month-old male and female mice. Fibres in the culture were under different treatments. 6-NBDG incubation for 19 h. Treatments exposure was 20 h: insulin, hydrogen peroxide, and angiotensin II. Data from individual fibres cultured in independent plates are presented as the means of net 6-NBDG intracellular fluorescence, number of fibres (n), and SEM. Statistical analysis: one-way ANOVA. Results were not statistically significant based on ANOVA (*p* > 0.05). Thus, the post-hoc test for paired comparisons was un-applicable. Light transmission and fluorescence emission microscopy images show individual fibres representative of every experimental condition.

**Figure 10 ijms-24-08082-f010:**
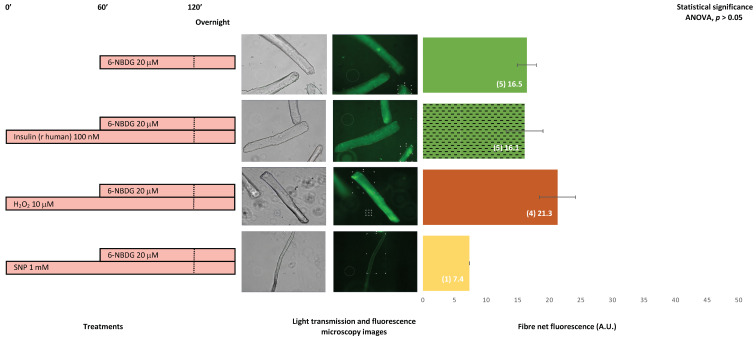
6-NBDG uptake in skeletal muscle fibres isolated from 25–29-month-old male mice. Fibres in the culture were subjected to different treatments. 6-NBDG incubation for 19 h. Treatments exposure was 20 h: Insulin, hydrogen peroxide, and SNP. Data from individual fibres cultured in independent plates are presented as the means of net 6-NBDG intracellular fluorescence, number of fibres (n), and SEM. Statistical analysis: one-way ANOVA. No statistical significance was observed based on ANOVA (*p* > 0.05); thus, the post-hoc test for paired comparisons was un-applicable. Light transmission and fluorescence emission microscopy images show individual fibres representative of every experimental condition.

## Data Availability

Not applicable.

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
