# Peer review of "Effect of RONS-Induced Intracellular Redox Homeostasis in 6-NBDG/Glucose Uptake in C2C12 Myotubes and Single Isolated Skeletal Muscle Fibres"

_ijms, 2023, doi:10.3390/ijms24098082_

Round 1
Reviewer 1 Report (New Reviewer)
Thank you for the invitation to review this manuscript. I have one major point, which is currently preventing me from adequately analysing the data.
The figures are not remotely acceptable for publication, in the current format they are largely unintelligible. The cell images are compressed to the point of being largely meaningless. Can I please ask the authors to present the data in a more conventional manner? If that can be achieved, then I evaluate the manuscript in its entirety.
Author Response
Please, see the attachment.

Reviewer 2 Report (New Reviewer)
lines 74-95: Is 6-NBDG different from 2-NBDG? Does 6-NBDG offer better characteristics for measuring glucose uptake? Please develop this topic in the text, taking into consideration that 2-NBDG uptake has been studied in muscle cells and isolated fibers in numerous publications
Figures 1 to 5: For the figures depicting myotubes (figures 1 to 5), the design is not ideal. The images of the myotubes are too small to clearly identify any differences in fluorescence. Additionally, the chosen method to display statistical differences is confusing
Experiments with H2O2: Was potential cell death evaluated when using H2O2 in myotubes and fibers? Were the concentrations of H2O2 used physiological or even pathological?
Figure legends: The first sentences in all figure legends are confusing. For example, in figure 1.1, the legend states "6-NBDG uptake in C2C12 myotubes under different treatments during 2 h and 6-NBDG for 1 h: insulin (bovine origin), hydrogen peroxide, and insulin (bovine origin) and hydrogen peroxide." It is not clear whether the timeframe of incubation is 2 hours or 1 hour. This applies to all figure legends.
In experiments overnight: How physiological is it to expose cells to 19 hours of treatment, such as 19 hours of hyperinsulinemia? Did the overnight treatments cause cell death?
lines 214-216: Authors mentioned that DEM and diamide caused toxicity in myotubes. How that toxicity was measured? GSHEE pre-treatment prevented that toxicity? How much?
Figure 4: In figure 4 insulin failed to induce 6-NBDG uptake. Why?
Figure 6: Once again, insulin failed to induce 6-NBDG uptake; however, the accompanying image of fibers appears to show increased fluorescence
lines 288-289: Please note that a 19-month old mouse is considered an aged mouse
Figure 8: Authors mentioned that in 25-29 month old mice, only a few fibers were viable, and even in fibers treated with SNP, there was only one viable fiber with atypical morphology. I recommend increasing the data collection to ensure reliable conclusions
lines 345-361: Authors suggest that 2-NBDG may be metabolized inside the cell, unlike 6-NBDG. Could you please discuss more this idea and provide evidence to support this statement? Do you have data on the use of 2-NBDG after overnight incubation?
lines 380-383: Please provide arguments for that statement
lines 399-402: "We observed that myotubes exposed to H2O2 presented similar glucose uptake to the control myotubes and found that glucose uptake improved when myotubes were incubated with H2O2 and insulin concomitantly" At all H2O2 concentrations used?
lines 403-406: "Moreover, it appears that there might be an interaction between insulin and hydrogen peroxide, since when hydrogen peroxide and insulin are concomitantly present, the glucose uptake increase evoked by insulin is attenuated" The discussion from line 399 to line 412 is confusing. Please clarify in which context H2O2 improved insulin-mediated 6-NBDG uptake and in which context H2O2 interfered with the uptake.
Through all the discussion, it is mentioned that oxidative stress enhances 6-NBDG uptake in some contexts while inhibiting it in others. Please discuss this apparently dual effect of reactive oxygen species
lines 496-513: So, if part of 6-NBDG can enter the cell through GLUT-1 transporters, how could this effect be determined? Considering that although insulin can induce significant increases in 6-NBDG uptake, they are not of great magnitude compared to the control. Therefore, there could be a significant interference of uptake mediated by GLUT-1.
Author Response
Please, see the attachment.

Reviewer 3 Report (Previous Reviewer 1)
As mentioned in my previous report the authors did address the majority of concerns raised and modified the manuscript accordingly. I do not have any objections regarding the manuscript getting published, provided that the authors have addressed the issues raised by other reviewers and some minor typos have been corrected in the manuscript.
Author Response
Please, see the attachment.

Round 2
Reviewer 2 Report (New Reviewer)
In the supplementary figures, please show only the images of myotubes and muscle fibers (without the bars) in an appropriate resolution. It is important to have a proper visualization of cells.
What does it mean when there are dashed lines that are darker than others? Please clarify the way of showing significant differences because it is still confusing
After reading the answer 4, choose another way to express it in the legends for greater clarity
Answer 6: Please add to the main text the method described in answer 6 for evaluating toxicity in the cells
Answer 8: Please change the image to one that is more appropriate for the behavior of the group of fibers.
Answer 9: https://www.jax.org/news-and-insights/jax-blog/2017/november/when-are-mice-considered-old
Answer 10 : Please, do not display results without having the appropriate number of replicates
Answer 12: Incorporate what was discussed in answer 12 into the main text
Author Response
Please, see the attachment.

This manuscript is a resubmission of an earlier submission. The following is a list of the peer review reports and author responses from that submission.
Round 1
Reviewer 1 Report
This manuscript introduces a new methodology/tool to determine glucose uptake and thereby investigates the effect of RONS and intracellular redox homeostasis in skeletal muscle. This article builds upon many previous studies that used fluorescence glucose analogs to study glucose uptake. The author has used 6-NBDG, a fluorescent glucose analog that does not undergo any transformation and metabolization upon transportation across the plasma membrane, and claims to be the first to use it to determine glucose uptake in C2C12 myotubes and single isolated skeletal muscle fibers. Overall, the data looks quite straightforward; the paper is simple in concept and the figures generally support the conclusions drawn. After addressing the following issues, this paper should be appropriate for publication.
The authors might consider changing the title pertaining to the discovery of a new method for determining glucose uptake rather than the effect of RONS in glucose uptake since there are many previous studies linking ROS and nitric oxide with glucose uptake.
Please give reasons for isolating single skeletal muscle fibers from the flexor digitorum brevis mouse muscle and not from any other muscle like the gastrocnemius which is also studied for investigating metabolic diseases such as diabetes and obesity. Are there any studies that show FDB mouse muscle to be the most appropriate to study glucose uptake, please cite them.
The authors are suggested to provide negative controls for figures 1.2.1, 1.2.2, 2, 3, 4 and 5 for comparing the fluorescence level with different treatment groups and eliminating the effect of autofluorescence.
The authors are suggested to provide justification in the discussion for the reduced fluorescence/glucose uptake observed in the combined effect of insulin and hydrogen peroxide or H2O2 alone.
In figure 2 the author may remove the results for Angiotensin II (10μM) which shows increased glucose uptake which contradicts previous reports that mention Angiotensin II to decrease glucose uptake (PMID: 2038279, PMID: 16982630). Similarly in figure 7, in skeletal muscle fibers, it is shown that Angiotensin II favors glucose uptake which again is contrary to the above-mentioned reports.
Although the authors have tried to justify that they have shown the effect in the absence of insulin and the increase in glucose uptake in the case of Angiotensin II treatment may occur through the AMPK signaling pathway, they do not have any experimental data to prove it. So, it is suggested that if possible the authors should show the effect of Angiotensin II in the presence of insulin on the glucose uptake in C2C12 myotubes.
The authors are suggested to repeat the experiments showing the effect of nitric oxide on glucose uptake with NOC, SNP or any other nitric oxide donor to relate the results observed while using SNAP in C2C12 myotubes.
Minor Correction:
The authors are suggested to correct the grammatical and spelling errors throughout the manuscript.
Please edit or correct the concentration units (μM) in the figures.
The authors should correct the results in section 2.3 where they have written MEM instead of DEM.
In ‘Experimental Procedure’ cytochalasin B has been misspelled.
The authors can describe the skeletal muscle fiber isolation method in more detail.
Author Response
Please, see the attachment

Reviewer 2 Report
The authors attempted to develop a methodology to detect glucose uptake based on the fluorescence glucose analogue 6-NBDG in C2C12 myotubes and single muscle fibers. Authors reported reactive oxygen and nitrogen species (RONS) play an important role in glucose uptake using this method. There are also significant findings, such as observing the effects of aging on glucose uptake into single fibers. However, it is difficult to determine if glucose is measured using the methodology authors developed in this paper. Listed below are my specific comments.
Major comments
・At first, it is necessary to prove that glucose is measured by this method. For example, the author needs to clarify that the fluorescence intensities change depending on the glucose concentration in media when the C2C12 myotubes and single muscle fibers were cultured in media with different glucose concentrations.
Results
・The result is a list of results, and it is not clear why each experiment was conducted. Authors should describe the reasons why they used this drug combination (for example insulin and H2O2). I do not understand why they used this drug combination.
・In some results, it is described as an increase or decrease even though the difference is not significant (line283, 284, 306). Results should be written according to statistical results.
・Throughout the manuscript, what is the toxicity? It must be shown quantitatively. It is necessary to show the number of living cells by the number of tubes and/or to show the number of cell viability by the assay in the results.
・There is data for one fiber (n=1) (Figure 8). It is not enough to obtain the conclusion for the scientific paper.
Methods
・Authors should describe the analysis method in more detail. How many dishes were performed in each condition. How many images were acquired per 1 dish. How many myotubes were analyzed.
・Typicaly, the quantification of glucose uptake by insulin is not performed overnight. An hour or tens of minutes after adding insulin would be appropriate.
Author Response
Please, see the attachment.

Round 2
Reviewer 1 Report
The response by the authors to the concern raised has been properly addressed and the majority of the suggestion has been incorporated in the revised manuscript. I do not find any issue with the revised manuscript considering that some of the experiments suggested will be addressed by the authors in their future studies and that doing the experiments at this stage of publication is not possible as mentioned by the authors. Nonetheless, I think the revised manuscript can be accepted in its present form in the International Journal of Molecular Sciences.
Author Response
Please, see the attachment.

Reviewer 2 Report
I cannot guarantee that glucose uptake can be measured by this method. The authors postulate that GLUT4 translocation controls uptake in C2C12 myotubes. I disagree with this idea. First, C2C12 is rarely used to study insulin-dependent glucose transport. C2C12 cells do not express sufficient levels of GLUT4 proteins, and GLUT4 translocation is reportedly minimal even after differentiation (PMID: 8463552; PMID: 1584210). Second, C2C12 has very LOW GLUT4 and HIGH GLUT1 protein contents (PMID: 25616865). Since GLUT1 is constantly present in the plasma membrane, glucose uptake is thought to increase in a glucose concentration-dependent manner in media as proposed in the first review comments. It is unclear whether this method can measure glucose uptake because of the above. It is necessary to measure glucose uptake by a biochemical method using 2DG glucose, at least, and demonstrate a positive correlation with glucose uptake measured by this method.
Author Response
Please, see the attachment.
